# A Generalized Convolutional Neural Network for Small Dataset Classification

## Abstract

We propose a novel variant of neural networks, Generalized Convolutional Neural Networks, GConvNets, characterized by structured neurons. In contrast to conventional neural networks such as ConvNets, which predominantly employ 'scalar' neurons, GConvNets utilize structured 'tensor' neurons. In other words, we generalize ConvNets by substituting each scalar neuron in ConvNets with a tensor neuron in GConvNets, while preserving the weight-sharing mechanism. These structured neurons manifest as tensors with adaptable shapes and dimensions across different layers. To ensure their practical applicability, we have developed a mechanism that enables seamless handling of hybrid structured tensor neurons as they transition from one layer to the next. We conducted a comparative analysis between GConvNets and the currently popular ConvNets, which include ResNets, MobileNets, EfficientNets, RegNets, among others, using datasets such as CIFAR10, CIFAR100, and Tiny ImageNet. The experimental results demonstrate that GConvNets exhibit superior efficiency in terms of parameter usage.

## 1 Introduction

ConvNets are continually growing in size. A notable example is ChatGPT, which comprises an impressive 175 billion parameters—twice as many as the human brain. While the initial motivation for the increasing size of models was primarily driven by their improved performance on large datasets, subsequent research has revealed additional advantages associated with these larger models. One of them is discriminability. With an increased number of parameters, larger models have the potential to capture intricate and complex features when provided with ample data and training epochs, leading to improved performance on challenging tasks. Recent advancements in generative models for languages, images, and videos further substantiate this claim. To summarize, the preference for larger model sizes stems from their ability to exhibit superior performance on large datasets, achieve commendable results on complex tasks, and enhance robustness when confronted with adversarial scenarios.

### 1.1 Overparameterization & traditional solutions

Nowadays, large models often comprise billions of parameters, which limits their usage in resource-constrained devices or real-time applications. Consequently, researchers have been reflecting the necessity of using so many parameters. This contemplation has revealed that ConvNets often suffer from the problem of overparameterization. In response, researchers have devised various approaches to reduce the size of ConvNets, including pruning, compression, distillation, binarization, and others. Pruning involves the removal of connections or neurons in a network that has limited contribution to the final output or limited impact on performance. This can be accomplished by directly removing connections with small weights or activations, or by applying heuristic algorithms to selectively remove neurons to achieve a balance between maximizing neuron removal and preventing significant degradation in the pruned model's performance. Compression generally adopts matrix decomposition techniques like SVD or Tucker decomposition, etc. to replace the original weight matrix with smaller ones. Binaryzation replaces float weights with binary numbers, which may reduce a model's size at the cost of performance. Distillation has the classic teacher-student framework, where a smaller neural network (the student) is trained using soft targets provided by the original network (the teacher). As a result, the knowledge is transferred from the teacher net-

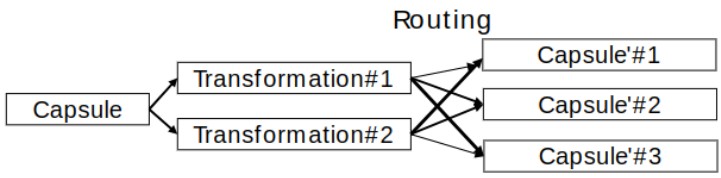

Figure 1: Capsule Networks.

work to the student network, resulting in the student achieving comparable performance while being more compact. Each of these techniques may have varying degrees of impact on the performance compared to the original model.

All of the mentioned approaches follow a common principle: initially constructing large ConvNets and subsequently reducing their sizes. By employing pruning and compression techniques, it is possible to reduce the size of many large ConvNets, sometimes up to 90%. This raises the question: if a significant portion of parameters in ConvNets can be reduced, why not utilize fewer parameters from the beginning? Can we create a compact neural network in the first place? Some researchers have discovered that traditional ConvNets with some structured neurons or hidden representations help alleviate overparameterizations. Structured neurons or hidden representations mean that we consider a group of neurons or the corresponding activations as a whole, where they together serve one target.

## 1.2 STRUCTURED HIDDEN REPRESENTATIONS

One example of such an approach is Capsule Networks (CapsNets). CapsNets consist of convolutional layers, capsule layers, and capsule output layers. The convolutional layers have the same structures as those in ConvNets, and so do their roles (extracting low-level features like edges, colors, etc.) In capsule layers, the neurons are re-grouped as capsules. The output of each capsule layer is a set of vectorized hidden representations, with each vector representing the presence or property of an entity. These hierarchical structures built by capsules are supposed to capture the relationship between different entities at various levels. Thus, CapsNets as a whole can better model spatial relationships. Furthermore, CapsNets incorporate design elements such as routing mechanisms (which determine coupling coefficients between capsules in adjacent layers) and vectorized output. However, the key innovation in CapsNets lies in the concept of capsules. Despite their primary purpose of capturing spatial relationships and handling viewpoint variations, the hierarchical nature of CapsNets allows for fewer neurons (parameters) to achieve the same or better performance (Hinton et al., 2018).

Another example is the attention mechanism. It draws inspiration from the way human attention operates, allowing individuals to dynamically monitor various information while simultaneously handling multiple tasks. A typical attention mechanism involves three vectors, namely query, key, and value. The hierarchical structures of the attention mechanism make it fall into the slot of structured hidden representations. Similar to the routing mechanism in CapsNets, the attention mechanism also assign weights to different parts of the input sequence based on similarities. Some popular neural network structures like Transformers are also built on top of attention mechanisms. Networks usually achieve better performance and efficiency (fewer parameters) after introducing attention mechanisms.

In summary, we can observe two similarities between the attention mechanism and CapsNets. Firstly, both methods incorporate structured hidden representations into ConvNets. Secondly, they both utilize information routing mechanisms to assign varying degrees of importance to different elements or entities in the input data.

## 1.3 STRUCTURED NEURONS

It is worth noting that both Transformers and CapsNets leverage feature maps originating from convolutional layers and subsequently re-organize these feature maps. Why not produce these structured

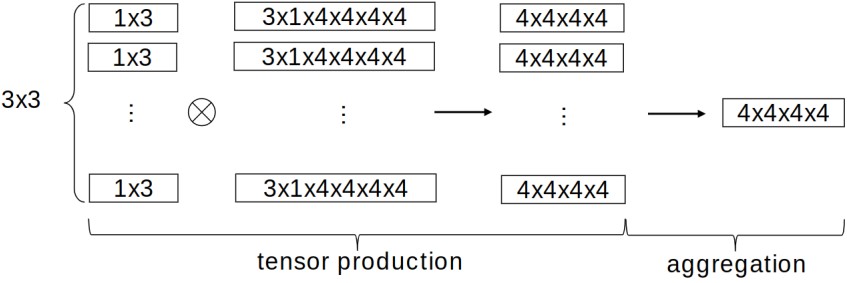

Figure 2: An example of Generalized Convolution. The 3x3 structured hidden representations, with a shape of 1x3x3x(1x3), transform to yield a single structured tensor with a shape of 1x(4x4x4x4). Note that in this context, the term 3x3 refers to the kernel size, and we make the assumption that both the input feature maps and output feature maps consist of only one channel (the 1s in the first dimension).

hidden representations from nerons directly? GConvNets employ structured neurons to directly generate these structured representations, eliminating the need for this additional step. Furthermore, GConvNets regard ConvNets as its special form, wherein each structured neuron comprises only a single scalar neuron. Consequently, we can seamlessly substitute the convolutional layers in ConvNets with general convolutional layers in GConvNets throughout the entire network. These structured neurons function as the fundamental building blocks across the network, diverging from the specialized components observed in Transformers and CapsNets.

### 1.4 GENERALIZED CONVNETS BUILT ON STRUCTURED NEURONS

So how are we supposed to build such a neural network? First, we employ high-rank tensors as the structured neurons and utilize the tensor product as the fundamental operations within the network. Secondly, we eliminate the need for information routing procedures, as the importance-based attention or coupling coefficients are learnable. By removing the information routing procedures, we can significantly reduce the overall overhead involved in the process. Next, we incorporate parameter sharing within each layer, similar to how convolutions operate, but with the use of structured neurons. However, it is important to note that CapsNets or the attention mechanism do not involve parameter sharing, and instead follow a similar style to fully-connected neural networks.

In summary, we adopt high-rank tensors as structured neurons, employ parameter sharing within each layer similar to ConvNets, package and unpackage the input/output to form and decompose tensors, maintain the same initialization, activation, and loss functions, and make the tensor product the default operation across layers. This new neural network, known as Generalized Convolutional Neural Networks (GConvNets), follows a similar organizational structure to ConvNets, except that each neuron is replaced with a high-rank neuron tensor. Therefore, when each structured neuron comprises only a single scalar neuron, GConvNets essentially reduce to regular ConvNets. In other words, ConvNets can be viewed as a specialized type of GConvNets, while GConvNets represent a more general form of ConvNets. Figure 2 shows one example of generalized convolutions.

What advantages do GConvNets offer? First, GConvNets offer greater flexibility when it comes to designing neural networks. In ConvNets, all layers consist of convolutional layers with scalar neurons. In contrast, GConvNets allow for the utilization of diverse tensor neurons with varying shapes and dimensions in different layers. Furthermore, our experiments demonstrate that employing high-rank neuron tensors can lead to more efficient model construction, requiring fewer parameters.

## 2 RELATED WORK

GConvNets adopt structured neurons to produce structured hidden representations. Similarly, CapsNets (Hinton et al., 2018) and Transformers (Vaswani et al., 2017) also take advatange structured hidden representations by reorganizing extracted features.

CapsNets (Sabour et al., 2017) organize neurons as capsules to mimic the biological neural systems. Different from normal neural networks, which adopt neurons as basic units, CapsNets use a group of neurons as capsules. A typical CapsNet is composed of several convolutional layers, a final fully-connected capsule layer with a routing procedure, and a loss function. Another key design of CapsNets is the routing procedure which can combine lower-level features with higher-level features to better model hierarchical relationships. CapsNets can encode intrinsic spatial relationships among features (parts or a whole) more efficiently than ConvNets. For example, the CapsNet with dynamic routing (Sabour et al., 2017) can separate overlapping digits accurately, while the CapsNet with EM routing (Hinton et al., 2018) achieves a lower error rate on smallNORB (LeCun et al., 2004). In contrast, ConvNets are usually overparameterized. As shown in (Liebenwein et al., 2020; Yang et al., 2020; Li et al., 2020; Singh & Alistarh, 2020; van Baalen et al., 2020), their compressed/pruned neural networks have much smaller sizes with hardly any accuracy drop. As a result, CapsNets usually need a lot fewer parameters when reaching the same accuracy.

Transforms typically use a multi-headed attention mechanism. The assumption is that each attention head has a separate projection of the representations, and multi-head attention can thus take advantage of multiple representations subspaces. The representations are composed of $(key, value, query)$ triplets. In particular, each triplet contains three matrices $(K, Q, V)$. A linear transformation is applied between representations in adjacent layers, as Equation 1 shows,

$$att_i\left(K_i, Q_i, V_i\right) = softmax\left(\frac{Q_i K_i^T}{d_i}\right) V_i \tag{1}$$

Where $d_i$ is the length is $K_i$. When the attention heads are stacked and transformed linearly, we get the values of a multi-head attention, as Equation 2 shows,

$$multi\_att\left(K, Q, V\right) = [att_0, stt_1, \ldots att_n]W \tag{2}$$

Where $W$ is the linear transformation matrix after the attention heads are stacked on top of each other. Fundamentally, the representations of a higher layer are weighted combinations of the representations in the lower layer. The weights are calculated based on the similarities between queries in the higher layer and keys in the lower layer.

We can see here both the matrix (vector) capsules and the key/query vectors in transformers encode representations in a structured way. GConvNets differentiate from ConvNets with structured hidden representations in three key aspects:

- GConvNets employ structured neurons to directly generate these structured hidden representations, whereas Transformers or CapsNets produce these structured representations by rearranging the feature maps from ConvNets.

- Structured neurons serve as the fundamental units throughout the entire GConvNets, as opposed to being specialized components integrated into ConvNets.

- GConvNets do not rely on information routing mechanisms or specific encoding algorithms. The relationships between adjacent structured hidden representations in GConvNets are learnable, enabling the model to capture the necessary information without predefined procedures.

## 3 GConvNets

The primary operation at the core of ConvNets is convolution, which is a linear combination consisting of two sequential steps. Initially, it involves performing an element-wise multiplication between scalar neurons of the kernel size and feature maps of the same size. Following this, the resultant scalar values are aggregated to yield a single scalar value in the subsequent layer. In contrast, the foundational operation in GConvNets, known as generalized convolution, entails a linear combination of tensors. Unlike traditional convolutions, generalized convolutions commence by conducting tensor product operations between structured neurons of the kernel size and structured hidden representations of the same size. Afterward, a combination operation is applied to these output tensors, which are structured hidden representations.

As a consequence, every convolution operation yields a single value, whereas each step of generalized convolution generates a multidimensional tensor. This leads to ConvNets and GConvNets

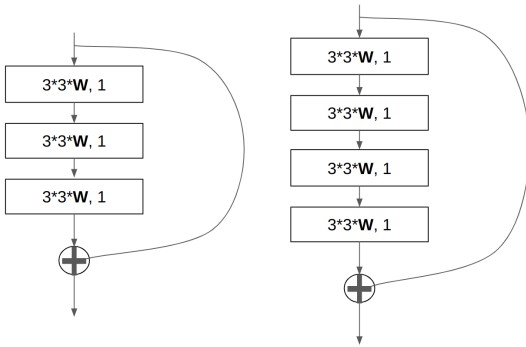

Figure 3: Two GConvNets residual structures. Left: the triple skips block, **Block#1** Right: the quadruple skips block, **Block#2**. $W \in \mathbb{R}^{k_1 \times k_2 \times k_3 \times k_4}$ is the neuron tensor. Each layer is followed by a PReLU (He et al., 2015) layer and a Batch Normalizaiton Layer (Ioffe & Szegedy, 2015)

having analogous yet distinct types of feature maps. Standard feature maps maintain a consistent homogeneity across layers, primarily differing in their dimensions (height, width, and number). Conversely, feature maps within GConvNets can display varying structures across layers. For example, a ConvNet feature map might be denoted as 32x128x128, signifying the presence of 32 channels, each with dimensions 128x128. In contrast, GConvNet feature maps are expressed as 32x128x128xT, where T can vary between layers, ranging from T=1 to more complex tensors like T=3x4x5x6. When T equals 1, a generalized convolutional layer essentially becomes a conventional convolutional layer. Moreover, if T remains consistently equal to 1 throughout the network, GConvNets essentially transform into ConvNets.

Next, we provide a comprehensive explanation of generalized convolution and elaborate on certain adjustments we implement to ensure the feasibility of training GConvNets.

### 3.1 GENERALIZED CONVOLUTION

Structured neurons serve as the fundamental units within the entire GConvNet architecture, responsible for converting input feature maps into output feature maps. For instance, they can transform an input tensor, denoted as $\mathbf{U} \in \mathbb{R}^{1 \times 2 \times 3 \times 4}$, into an output tensor, denoted as $\mathbf{V} \in \mathbb{R}^{1 \times 2 \times 7 \times 8}$, through the application of a tensor product operation with $\mathbf{W} \in \mathbb{R}^{4 \times 3 \times 7 \times 8}$, as Equation 3 shows. In theory, the structured neurons within GConvNets have the capability to convert tensors of any shape into tensors of any other shape. This particular step serves a role analogous to the head projection + attention distribution in Transformers or the capsule transformation + routing in CapsNets.

$$\mathbf{V}_i = \mathbf{U}_i \bigotimes \mathbf{W}_i \tag{3}$$

After the tensor production, GConvNets apply liner combinations, which can be defined as,

$$\mathbf{V} = \sum_i^n \mathbf{V}_i \tag{4}$$

Where $n = k \times k \times m$, $k$ is the kernel size and $m$ is the number of input channels. In comparison, the basic operation of ConvNets is linear combinations of scalars, $V = \sum_i^n V_i$.

### 3.2 INPUT&OUTPUT IN GCONVNETS

We process each input image as in Figure 2 shows. To illustrate, consider a 128x128x3 color image, which can be regarded as a one-channel structured feature map with dimensions of $\mathbf{U} \in \mathbb{R}^{1 \times 128 \times 128 \times (1 \times 3)}$. If we employ a structured neuron with dimensions of ($\mathbf{W} \in \mathbb{R}^{(3 \times 3) \times 1 \times 1 \times 3 \times 1 \times 4 \times 4 \times 4 \times 4}$), the input can be transformed to a one-channel structured feature map with dimensions of $\mathbf{V} \in \mathbb{R}^{4 \times 63 \times 63 \times (4 \times 4 \times 4 \times 4)}$, as Equation 3 shows. In this context, we are assuming a kernel size of 3, a stride of 2, and no padding.

Table 1: The structure of **g_conv_net**. The neuron tensor of *Layer#0* is $\mathbf{W}^{in} \in \mathbb{R}^{3 \times 1 \times 9 \times 9}$ that can transforms each input tensor $\mathbf{U}^{in} \in \mathbb{R}^{3 \times 1}$ to an output tensor $\mathbf{V} \in \mathbb{R}^{9 \times 9}$. *Layer#0* is followed by a BatchNorm Layer and a PReLU layer. So are the following layers. In the immediate layers, $\mathbf{W}^{in} \in \mathbb{R}^{9 \times 9 \times 9 \times 9}$ that can transforms each tensor $\mathbf{U}^{in} \in \mathbb{R}^{9 \times 9}$ to an output tensor $\mathbf{V} \in \mathbb{R}^{9 \times 9}$. In *Layer#4*, $\mathbf{W}^{f} \in \mathbb{R}^{4 \times 9 \times 9 \times 9 \times 9}$. In *Layer#6*, the final output number depends on the number of classes.

| #Layer | Neural Tensors | #Channel |
|:---:|:---:|:---:|
| 0 | **Block#1** | 1 |
| 1 | **Block#2** | 1 |
| 2 | **Block#1** | 1 |
| 3 | **Block#2** | 1 |
| 4 | $3 \times 3 \times \mathbf{W}^{f}$ | 4 |
| 5 | Pooling | 4 |
| 6 | $324 \times 10/100/200$ | 10/100/200 |

With both the input and the neurons structured, the resulting output naturally inherits a structured format. Consequently, this necessitates the loss functions specifically designed to handle structured data, such as the margin loss in CapsNets. However, these loss functions often require additional steps. A more convenient approach is to destructurize the output and then use a regular loss function. Specifically, we deconstruct the structured feature maps into normal feature maps, subsequently incorporating a pooling layer prior to inputting them into a loss function.

To streamline the design process and leverage the advancements made in ConvNets, we preserve most other components in ConvNets, including activation functions, initialization functions, batch normalizations, loss functions, residual structures, etc., untouched. The residual structures in our paper can be found in Figure 3.

## 4 EXPERIMENTS

We conducted experiments with GConvNets on several small datasets, specifically CI-FAR10 (Krizhevsky et al.), CIFAR100 (Krizhevsky et al.), and TinyImageNet (Le & Yang, 2015). Initially, we evaluated GConvNets against other ConvNets in a conventional manner, employing various data augmentation techniques such as resizing, cropping, flipping, and normalization. Subsequently, we conducted experiments without applying these techniques, maintaining the original input size and avoiding resizing, cropping, or flipping. Additionally, we omitted the normalization step based on prior knowledge. This approach was chosen to eliminate extraneous factors and provide a more accurate assessment and comparison of the performance across different neural network architectures. Throughout this paper, we maintain the same choice of a learning rate of 1e-3, a mini-batch size of 32, and the utilization of the Adam optimizer (Kingma & Ba, 2015) across all datasets and settings.

We employ a singular GConvNets, referred to as g_conv_net, for all the datasets. The comprehensive configuration of this network can be observed in Table 1.

### 4.1 CIFAR10

The CIFAR-10 dataset consists of 60000 32x32 color images in 10 classes, with 6000 images per class. There are 50000 training images and 10000 test images. For every sample in CIFAR10, our preprocessing pipeline involves resizing each image to dimensions of 42x42. Subsequently, we apply random cropping, resulting in a final size of 38x38. We also incorporate horizontal flipping for data augmentation purposes. Finally, each channel of an image is normalized using mean values of (0.485, 0.456, 0.406) and standard deviations of (0.229, 0.224, 0.225).

We then refrain from any preprocessing of the original CIFAR10 data. Specifically, we utilize the original input size of 32x32 and do not apply resizing, cropping, or normalization. As Table 2 shows,

Table 2: GConvNets Performance Results on CIFAR10, CIFAR100, and Tiny ImageNet. The epoch# here is the epoch number when the lowest validation loss is recorded. For each accuracy item, the number on the left indicates the utilization of augmentations, while the number on the right indicates the absence of any augmentations.

| Datasets | Model | Acc. | #Params | #Epochs |
|---|---|---|---|---|
| CIFAR10 | shufflenet_v2_x0_5 (Zhang et al., 2018) | 76.5%/60.5% | **0.35M** | 47/9 |
| | **g_conv_net** | **88.4%/ 78.4%** | 1.43M | 71/11 |
| | mobilenet_v3_small (Sandler et al., 2018) | 80.0%/63.2% | 1.53M | 17/13 |
| | mobilenet_v2 (Sandler et al., 2018) | 84.4%/73.0% | 2.24M | 32/17 |
| | regnet_y_400mf (Radosavovic et al., 2020) | 82.7%/68.0% | 3.91M | 30/14 |
| | EfficientNet-B0 (Agarwal et al., 2019) | 85.6%/70.0% | 4.0M | 79/8 |
| | regnet_y_800mf (Radosavovic et al., 2020) | 82.9%/33.5% | 5.66M | 21/6 |
| | EfficientNet-B1 (Agarwal et al., 2019) | 87.2%/72.9% | 6.5M | 66/14 |
| | EfficientNet-B2 (Agarwal et al., 2019) | 86.1%/75.0% | 7.72M | 43/20 |
| | resnet18 (He et al., 2015) | 86.1%/73.9% | 11.18M | 13/6 |
| | resnet34 (He et al., 2015) | 86.5%/74.2% | 21.29M | 17/8 |
| | convnext_tiny (Liu et al., 2022) | 76.2%/60.4% | 27.82M | 19/5 |
| | convnext_small (Liu et al., 2022) | 84.6%/59.8% | 49.45M | 15/5 |
| CIFAR100 | shufflenet_v2_x0_5 (Zhang et al., 2018) | 39.7%/32.3% | **0.44M** | 14/14 |
| | **g_conv_net** | **56.8%/44.4%** | 1.46M | 81/23 |
| | mobilenet_v3_small (Sandler et al., 2018) | 37.2%/33.8% | 1.62M | 20/28 |
| | mobilenet_v2 (Sandler et al., 2018) | 54.5%/38.8% | 2.35M | 46/20 |
| | regnet_y_400mf (Radosavovic et al., 2020) | 46.0%/31.9% | 3.95M | 19/8 |
| | EfficientNet-B0 (Agarwal et al., 2019) | 53.0%/35.0% | 4.14M | 38/21 |
| | regnet_y_800mf (Radosavovic et al., 2020) | 49.2%/38.2% | 5.73M | 13/10 |
| | EfficientNet-B1 (Agarwal et al., 2019) | 53.2%/32.2% | 6.64M | 80/8 |
| | EfficientNet-B2 (Agarwal et al., 2019) | 54.2%/44.3% | 7.84M | 13/14 |
| | resnet18 (He et al., 2015) | 53.8%/42.9% | 11.23M | 12/9 |
| | resnet34 (He et al., 2015) | 53.4%/42.4% | 21.34M | 17/7 |
| | convnext_tiny (Liu et al., 2022) | 42.8%/30.4% | 27.89M | 15/5 |
| | convnext_small (Liu et al., 2022) | 44.1%/32.6% | 49.52M | 16/8 |
| Tiny ImageNet | shufflenet_v2_x0_5 (Zhang et al., 2018) | 35.7%/28.6% | **0.55M** | 40/13 |
| | **g_conv_net** | **48.5%**/34.8% | 1.49M | 75/25 |
| | mobilenet_v3_small (Sandler et al., 2018) | 33.8%/28.7% | 1.72M | 83/18 |
| | mobilenet_v2 (Sandler et al., 2018) | 39.1%/32.4% | 2.35M | 35/16 |
| | regnet_y_400mf (Radosavovic et al., 2020) | 38.2%/29.3% | 3.99M | 21/12 |
| | EfficientNet-B0 (Agarwal et al., 2019) | 42.6%/32.4% | 4.26M | 90/13 |
| | regnet_y_800mf (Radosavovic et al., 2020) | 40.5%/**35.2%** | 5.8M | 18/8 |
| | EfficientNet-B1 (Agarwal et al., 2019) | 44.4%/32.2% | 6.77M | 92/17 |
| | EfficientNet-B2 (Agarwal et al., 2019) | 40.4%/31.3% | 7.98M | 109/16 |
| | resnet18 (He et al., 2015) | 40.0%/33.0% | 11.28M | 12/5 |
| | resnet34 (He et al., 2015) | 39.9%/34.2% | 21.39M | 19/5 |
| | convnext_tiny (Liu et al., 2022) | 37.7%/26.9% | 27.97M | 14/9 |
| | convnext_small (Liu et al., 2022) | 38.2%/27.6% | 49.59M | 14/9 |

g_conv_net achieves the best performance with/without data augmentations than most models using far fewer parameters.

## 4.2 CIFAR100 (KRIZHEVSKY ET AL.)

CIFAR100 has 100 classes containing 600 images each. There are 500 training images and 100 testing images per class. We first use exactly the same data augmentations as in section 4.1. Subsequently, we eliminate all augmentation techniques and re-evaluate our model. As evidenced by Table 2, g_conv_net exhibits the most superior performance among all models, both with and without data augmentations.

### 4.3 TINY IMAGENET (LE & YANG, 2015)

Tiny ImageNet (Le & Yang, 2015) is a subset of the ImageNet dataset (Deng et al., 2009), which contains 100,000 images of 200 classes (500 for each class) downsized to 64×64. The preprocessing pipeline encompasses the resizing of each image to dimensions of 80x80. Following this, random cropping is applied, resulting in a final size of 64x64. Additionally, horizontal flipping is incorporated for the purpose of data augmentation. Lastly, each channel of an image is normalized using mean values of (0.485, 0.456, 0.406) and standard deviations of (0.229, 0.224, 0.225).

Upon removing all preprocessing steps and retesting our model, we obtained results of 48.5% and 34.8% for the two respective settings. The utilization of data augmentation techniques leads to the highest performance for g_conv_net. Without augmentations, g_conv_net still achieves the second-best result with significantly fewer parameters compared to the best one.

### 4.4 DISCUSSION

One may wonder why GConvNets tend to outperform many larger ConvNets. We conjecture that GConvNets can better capture spatial features from the samples and thus be less vulnerable to over-fitting. The advantage that GConvNets exhibit is somewhat akin to the structured hidden representations employed by Transformers and CapsNets. However, it's important to note that GConvNets employ structured neurons to directly generate these structured hidden representations, whereas Transformers or CapsNets produce these structured representations by rearranging the feature maps from ConvNets.

The evidence supporting the reduced susceptibility to overfitting in GConvNets becomes apparent when observing their comparatively narrower generalization gaps. To illustrate, both ResNet34 (He et al., 2015) and EfficientNet-b3 (Agarwal et al., 2019) can attain nearly 100% accuracy on the Tiny ImageNet training set, yet their accuracy on the validation set remains consistently at around 30%.

This paper primarily concentrates on classification tasks involving small datasets. An intriguing inquiry is whether GConvNets can also surpass traditional ConvNets on larger datasets. Regrettably, due to the absence of hardware-accelerated algorithms, the training process for GConvNets is considerably slower when compared to ConvNets. Conventional convolution can be viewed as a small number of large matrix multiplications, whereas generalized convolution involves a significant number of small matrix multiplications, a process not optimally supported by the current GPU acceleration frameworks.

## 5 CONCLUSIONS

We propose a generalized variant of Convolutional Neural Networks named GConvNets, which employ structured neurons to generate structured hidden representations. In particular, we expand the traditional convolution operation from being a linear combination of scalars to involving tensor product and tensor aggregation, thus making convolution a specialized instance of generalized convolution. Furthermore, we have developed a range of techniques to empower GConvNets to harness the successful mechanisms utilized by ConvNets. Our experiments show that GConvNets demonstrate enhanced efficiency in classification tasks across various datasets. We believe that GConvNets have the potential to hold potential in other artificial intelligence tasks and datasets as well.

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
