# OpenReview forum: "A Generalized Convolutional Neural Network for Small Dataset Classification"
_ICLR.cc/2024/Conference — ICLR 2024 Conference Withdrawn Submission_

### Official Review · Reviewer_JBwA · 2023-11-01

**Soundness:** 2 fair
**Presentation:** 1 poor
**Contribution:** 1 poor
**Rating:** 3
**Confidence:** 3

**Summary:**

This paper proposes to use tensors instead of scalars from the beginning.  The authors create a compact network with structured neurons. Experimental results demonstrate the advantages of the proposed method.

**Strengths:**

In contrast to the basic operation of ConvNets where linear combinations of scalars is used, the authors proposed to replace scalars to tensors.  The idea is novel and the experimental results show advantage of the proposed methods.

**Weaknesses:**

1. The main concerns lies in the contribution. The authors merely replace scalars to tensors in the linear combination operation, and the contribution is limited.

2. The motivation is not clear and not convincing.

3. The writing and presentation can be improved.

4. More experiments should be conducted. For example,  different models' experimental results on the same network structures should be compared.

**Questions:**

Since the proposed method replace scalars to tensors, the parameter number should increase. I'm confusing why the parameter number decreases? How is your model improve performance?

---

> ### Author Response · Authors · 2023-11-13
>
> About the weaknesses:
> 1. I would say that GConvNets is more fundamental than many research papers. The concept might not be complex, but redefining scalar neurons as tensor neurons is anything but trivial. Consider this: ConvNets solely involve matrix multiplications between neurons and their corresponding features. In contrast, the potential operations between tensor neurons and structural representations are notably larger. We need to figure out the most efficient one from them. Furthermore, there are no well-supported acceleration algorithms for such operations, we have to develop our own. A further illustration is initialization. In ConvNets, the objective is to maintain a consistent variance of features across layers. However, when tensor productions enter the picture, the task becomes more intricate due to the necessity of addressing or simplifying the covariance within each tensor neuron. Additionally, we must develop tensor-based versions of Batch Normalization as well as other components in a neural network.
>
> 2. Our motivation is to propose a more general version of ConvNets, we can definitely make it more clear.
>
> 3. We agree with you that our presentation of our idea can be improved. Could you kindly elaborate on the aspects that require improvement?
>
> 4. I am a little bit confused, we use a single model to demonstrate GConvNets' efficiency across three datasets. While we could achieve better results by designing specialized models for each dataset, we believe this to be unnecessary. Maybe we misunderstood your idea, can you specify it?
>
> About the question, "Since the proposed method replace scalars to tensors, the parameter number should increase. I'm confusing why the parameter number decreases? How is your model improve performance?"
> In short, it's not merely replacing scalar neurons with structured neurons in ConvNets. Instead, when using structured neurons, the models can better grasp the spatial relationships within each image, resulting in fewer necessary parameters to achieve comparable performance.

---

### Official Review · Reviewer_wHyx · 2023-11-03

**Soundness:** 1 poor
**Presentation:** 1 poor
**Contribution:** 1 poor
**Rating:** 1
**Confidence:** 4

**Summary:**

Generalized Convolutional Neural Networks (GConvNets) are a proposed variant of neural networks with 'tensor' neurons instead of conventional 'scalar' neurons, allowing for adaptable tensor shapes and dimensions across different layers, while maintaining the weight-sharing feature.

**Strengths:**

Generalized Convolutional Neural Networks (GConvNets) are a proposed variant of neural networks with 'tensor' neurons instead of conventional 'scalar' neurons, allowing for adaptable tensor shapes and dimensions across different layers, while maintaining the weight-sharing feature.

**Weaknesses:**

1. Obvious mistakes. E.g. "ConvNets are continually growing in size. A notable example is ChatGPT". This lowers the quality and trustworthy of the paper.
2. 'Generalized CNN' seems to be overclaiming.
3. The core idea in Fig.2 is confusing. Based on understanding from the very limited information, it is just a tensor rewriting of the convolution and not novel.
4. why many figures/Eqn. from other works capsule and transformer network. They seem to have very little connection to the work presented here/

**Questions:**

1. Is it necessary to include Eqn. of transformers/attention in the related work?
2. " In contrast to conventional neural networks such as ConvNets, which predominantly employ ‘scalar’ neurons, GConvNets utilize structured ‘tensor’ neurons." I do not think this is a correct statement. Otherwise, explain 'scalar' defination.

---

> ### Author Response · Authors · 2023-11-13
>
> 1.Can you specify the "obvious mistake of "ConvNets are continually growing in size. A notable example is ChatGPT".?
>
> 2.We employ generalized ConvNets because the neuron tensor and the corresponding tensor production represent the general forms of scalar neuron and matrix multiplications.
>
> 3.GConvNets looks like replacing scalar neurons with tensor neurons. That' why we say it is more general. The concept might not be complex, but redefining scalar neurons as tensor neurons is anything but trivial. Consider this: ConvNets solely involve matrix multiplications between neurons and their corresponding features. In contrast, the potential operations between tensor neurons and structural representations are notably larger. We need to figure out the most efficient one from them. Furthermore, there are no well-supported acceleration algorithms for such operations, we have to develop our own. A further illustration is initialization. In ConvNets, the objective is to maintain a consistent variance of features across layers. However, when tensor productions enter the picture, the task becomes more intricate due to the necessity of addressing or simplifying the covariance within each tensor neuron. Additionally, we must develop tensor-based versions of Batch Normalization as well as other components in a neural network.
>
> 4.We believe that both GConvNets and CapsNets/Transformers use structured neurons/hidden representations, though the relation between them might be weak.
>
> 5. The scalar neuron is equivalent to those in ConvNets. While the structured neuron shares some similarities with capsules, it differs in that it employs high-rank tensors rather than matrices.

---

> > ### Comment · Reviewer_wHyx · 2023-11-22
> >
> > Thanks for your response. I still don't think this paper is ready for publication, so I decided to keep my original rating.
> >
> > in case you needed: 1) how can chatgpt be a convnet

---

> > > ### Author Response · Authors · 2023-11-23
> > >
> > > Thanks. Well, I still do not get it what makes chatgpt not a convolutional neural network.

---

### Official Review · Reviewer_8QHt · 2023-11-06

**Soundness:** 3 good
**Presentation:** 3 good
**Contribution:** 2 fair
**Rating:** 6
**Confidence:** 3

**Summary:**

The paper proposes a generalization of ConvNets using tensor processing and names the architecture GConvNets. GConvNets employ tensor operations per neuron compared to scalar operation employed by traditional ConvNets. The paper offers numerical studies on various datasets, comparing against various models. The paper claims that the superior performance of GConvNets is due to its ability to better capture spatial features from inputs and overcome over-parameterization. The paper is well written and has a decent flow. Please see below for detailed strengths and questions on the work.

**Strengths:**

The paper proposes an interesting idea of generalizing ConvNets using tensor processing. Experimental studies demonstrate the superiority of the proposed method compared to existing methods on classification performance and parameter efficiency. More interestingly, the paper claims that the plain convolutions are a spacial case of the proposed tensor processing.

**Weaknesses:**

Please see below.

**Questions:**

Please fix the following typos in the paper. I would encourage the authors to go over the paper thoroughly and fix any other typos.
1. Sentence after (1) needs to be modified. Currently it reads "Where $di$ is the length is $K$", which is not comprehendible.
2. Equation 2 contains a typo -- stt1.
3. The paper claims that the GConvNets perform better than ConvNets because they are able to better capture spatial features and are less vulnerable to overfitting -- although not vital, it would be great to see visualizations of feature maps to show how diffrent the features of GConvNets are, compared to ConvNets.
4. The paper mentions other works on parameter reduction in CNNs, including pruning, distillation, etc. but fails to cite relevant works or include them in comparisons. Could the authors cite relevant works in the section 1.1 and possible include comparisons with some of these methods (time permitting)?
5. Although the paper shows the #Params in the studies, it would be nice to see inference times also.

---

> ### Author Response · Authors · 2023-11-13
>
> 1. Thanks for pointing this out, we will modify this part.
> 2. Thanks! We will correct it.
> 3. This is a good suggestion. We visualized the neurons of the first two layers in GConvNets and the corresponding two layers in ConvNets, we found that the structured neurons in GConvNets are more closely correlated. We will give the result in our submission.
> 4. This is our fault, we will cite those works. Following your suggestion, We built a tiny version of our model and compared it with two compressed neural networks in [1] and [2]. We found that our model shows higher efficiency. In particular, our model contains around 22.2K parameters and achieves a 0.41 error rate on MNIST while the numbers of [1] and [2] are 52.5K/0.57 and 52.1K/0.51.
>
> 5. Regarding the speed of inference and training, we discovered that our current model versions are much slower than ConvNets. This issue is primarily attributed to the absence of acceleration APIs for tensor-based operations, such as in CUDA. We previously developed two CUDA-based acceleration algorithms in another deep learning framework, but they provided limited improvement. Theoretically, GConvNets should be on par with ConvNets in terms of training and inference speed as long as they have similar sizes, given that the main distinction lies in the number of parallel matrix multiplications.
>
> [1] Roger Baker Grosse and James Martens. A Kronecker-factored approximate Fisher matrix for convolution 256 layers. In International Conference on Machine Learning, 2016.
>
> [2] Z. Yang, M. Moczulski, M. Denil, N. d. Freitas, A. Smola, L. Song, and Z. Wang. Deep-fried convents. In 2015 IEEE International Conference on Computer Vision (ICCV), pages 1476–1483, Dec 2015. doi: 10.1109/ICCV.2015.173.

---

### Official Review · Reviewer_zai5 · 2023-11-07

**Soundness:** 2 fair
**Presentation:** 1 poor
**Contribution:** 2 fair
**Rating:** 5
**Confidence:** 2

**Summary:**

The paper presents a variant of neural networks, GConvNets for image classification. The major contribution is to use the so called  Generalized Convolutional to replace ordinary convolution. It employs high-rank tensors as the structured neurons and utilize the tensor product as the fundamental operations within the network. It uses parameter sharing within each layer, similar to ordinary convolution. Experiments on several image classification datasets show promising results of the GConvNet.

**Strengths:**

First, the idea is novel to me. Second, the results seem to be strong, though only on small datasets. The proposed network achieved high accuracy with much fewer parameters than existing methods including well-known ones such as EfficientNet and ConvNext.

**Weaknesses:**

First, the proposed method is not clearly described. I guess many readers in the AI community are not very much familiar with the tensor product with high ranks, and I'm one of them. Frankly speaking, I searched internet and still didn't get good answer that could match the description and examples well in the paper. It is strange that the paper spends much space for describing the well-known CapsuleNet and Transformer but does not describe the definition of the core of the proposed method clearly -- tensor product.

Second, the reason why the proposed method works well is not well explained. Throughout the paper, it is stressed that it is the "structured neurons" that bring the advantage of representation. CapsuleNet and Transformers are adopted as an analogy to explain this advantage. To me, this does not make much sense because the high rank tensor operations have little to do with the structures of the two models. BTW, it is hard to say Transformers have structured neurons.

Third, the efficiency is not reported in the results tables. In sec 4.4 it is stated that the proposed method works very slow. This is expected because the time complexity of tensor product is high. If efficiency is not resolved, the method is hardly to be recognized by the AI community.

In summary, the proposed method is a starting point of a potentially good work, but it is not ready to be published yet.

**Questions:**

Please see above 3 points.

---

> ### Author Response · Authors · 2023-11-13
>
> Thank you for your comment about the strength part. Our work does have this limitation. Although we believe that our approach might perform better than ConvNets in large-scale datasets, the absence of acceleration APIs for tensor-based operations in hardware acceleration frameworks like CUDA makes it challenging to validate this claim. We are currently working on an acceleration algorithm, but it may take some time to develop and implement. In the meantime, we are focusing on tiny datasets only.
>
> About the weaknesses,
> 1. Thanks for your feedback. We have removed some details about tensor production in this paper since we believed that they were more about math rather than neural networks, but we might have miscalculated. We can definitely add back these details.
> 2. I agree with you. We have given significant thought to how to present our concept, and we ultimately opted to use structured hidden representations/neurons to depict the similarity between GConvNets and Transformers/CapsNets, despite the correlation being relatively weak. At present, we believe that a more apt analogy would be the "hierarchical organization of neurons, which is akin to the human brain. Essentially, our focus is on explaining why tensor production is more efficient/general than matrix multiplication when it comes to mapping/projecting images to labels.
>
> 3. The efficiency in this paper is more about the number of parameters used rather than training/inference time. We discovered that our current model versions are much slower than ConvNets. This issue is primarily attributed to the absence of acceleration APIs for tensor-based operations, such as in CUDA. We previously developed two CUDA-based acceleration algorithms in another deep learning framework, but they provided limited improvement. Theoretically, GConvNets should be on par with ConvNets in terms of training and inference speed as long as they have similar sizes, given that the main distinction lies in the number of parallel matrix multiplications.

---

> > ### Comment · Reviewer_zai5 · 2023-11-22
> >
> > Thanks for your response. I still don't think this paper is ready for publication, so I decide to keep my original rating.

---

> > > ### Author Response · Authors · 2023-11-23
> > >
> > > Thanks!